# Advances of Hydroxyapatite Hybrid Organic Composite Used as Drug or Protein Carriers for Biomedical Applications: A Review

**DOI:** 10.3390/polym14050976

**Published:** 2022-02-28

**Authors:** Ssu-Meng Huang, Shih-Ming Liu, Chia-Ling Ko, Wen-Cheng Chen

**Affiliations:** 1Advanced Medical Devices and Composites Laboratory, Department of Fiber and Composite Materials, Feng Chia University, Taichung 407, Taiwan; dream161619192020@gmail.com (S.-M.H.); 0203home@gmail.com (S.-M.L.); rayko1024.rb@gmail.com (C.-L.K.); 2Department of Fragrance and Cosmetic Science, College of Pharmacy, Kaohsiung Medical University, Kaohsiung 807, Taiwan; 3Dental Medical Devices and Materials Research Center, College of Dental Medicine, Kaohsiung Medical University, Kaohsiung 807, Taiwan

**Keywords:** hydroxyapatite, nanoparticles, composites, drug release, protein, template, carriers, tissue engineering, scaffold

## Abstract

Hydroxyapatite (HA), especially in the form of HA nanoparticles (HANPs), has excellent bioactivity, biodegradability, and osteoconductivity and therefore has been widely used as a template or additives for drug delivery in clinical applications, such as dentistry and orthopedic repair. Due to the atomically anisotropic distribution on the preferred growth of HA crystals, especially the nanoscale rod-/whisker-like morphology, HA can generally be a good candidate for carrying a variety of substances. HA is biocompatible and suitable for medical applications, but most drugs carried by HANPs have an initial burst release. In the adsorption mechanism of HA as a carrier, specific surface area, pore size, and porosity are important factors that mainly affect the adsorption and release amounts. At present, many studies have developed HA as a drug carrier with targeted effect, porous structure, and high porosity. This review mainly discusses the influence of HA structures as a carrier on the adsorption and release of active molecules. It then focuses on the benefits and effects of different types of polymer-HA composites to re-examine the proteins/drugs carry and release behavior and related potential clinical applications. This literature survey can be divided into three main parts: 1. interaction and adsorption mechanism of HA and drugs; 2. advantages and application fields of HA/organic composites; 3. loading and drug release behavior of multifunctional HA composites in different environments. This work also presents the latest development and future prospects of HA as a drug carrier.

## 1. Introduction

Bioceramics are biocompatible ceramics, glass materials, or ceramic/glass composites designed to repair or rebuild damaged parts of human hard tissues [1]. For many decades of research, the focus has been on the mechanical properties and biocompatibility of bioceramics [2], while the current trend is toward functional polymer–bioceramic composites with additional therapeutic functions, such as antibacterial and antitumor [3]. Hydroxyapatite (HA) has a hexagonal crystal structure with the molecular formula Ca_10_(PO_4_)_6_(OH)_2_, and the structure of HA nanoparticles (HANPs) has two distinct binding sites. For example, HANPs have negatively charged phosphate anions at both ends and positively charged Ca^2^^+^ cations on the sides [4,5,6,7]. HA is an essential mineral for human bones, which consist of 70% low-crystalline or amorphous apatite, 30% collagen, and bone marrow cells [8,9,10,11,12]. At present, apatite preparation methods mainly include wet [13,14,15], dry [16,17,18], sol–gel [19,20,21], biological tissue synthesis [22,23,24], hydrothermal [25,26,27], and freeze-drying methods [28,29,30]. HA has good biological activity, biocompatibility, and non-toxicity. It is widely used in the filling and repair of hard tissues, such as bones and teeth [31,32,33]. When HA is mainly used as a carrier, especially in the form of HANPs, it can carry proteins, growth factors, antibiotics, anti-inflammatory drugs, tumor drugs, etc. [34,35,36,37], which can shorten the treatment time, achieve local sustained release, and guide tissue regeneration. The adsorption capacity of HANPs directly depends on their surface area, morphology, and hydration, which are usually regulated by pH and electrolyte concentration [38]. For drug delivery applications, the porosity and pore distribution in HANPs are important to determine drug loading capacity, drug delivery efficiency, and release kinetics [39]. Since charged and polar groups impart important properties to proteins through the formation of ion pairs, hydrogen bonds, and other less specific electrostatic interactions, the surface charge of HANPs can control protein binding through hydrogen bonding or electrostatic interactions, thereby providing binding sites between the protein molecules and the surface of HANPs [40]. Therefore, calcium cations (Ca^2^^+^) and phosphate anions (PO_4_^3−^) in HANPs can be used as preferential binding sites for proteins [41,42], the protein–mineral ion complex can be formed as a protein with specific ligand interactions; for example, peptides are inherently capable of binding Ca^2+^ to carbonyls by chelation.

HANPs are a calcium phosphate compound with similar composition to natural bone tissues and have excellent biocompatibility, osteoconductivity, osteoinductivity, and osteogenic ability; as such, HANPs are widely used in orthopedics and drug delivery systems [36,43,44,45]. The composite of HANPs and natural and synthetic polymers effectively solves the HANP problems, such as high brittleness, uncontrollable degradation rate, poor plasticity, and easy agglomeration [46]. Methods for compositing HANPs into a polymer matrix for processing include electrospinning, three-dimensional (3D) printing, freeze drying, etc. These hybrid composites can be formed into desired morphologies of braided thread, thin film, nanofiber composed membranes, scaffolds, microspheres or nano-beads, and sprayed coatings to enhance mechanical properties and adapt to target applications [47,48]. In addition to binding HANPs, these composites can be loaded with drugs, magnetic quantum dots, or grafted with growth factors or proteins according to different clinical needs. Depending on the resulting HANP composite polymers, these hybrid composites are widely used in various medicines, not limited to the regeneration of bone tissues.

## 2. HA as a Template for Protein/Drug Carriers

### 2.1. Carrier of HANPs for Protein Adsorption

For in vivo applications, nanoparticles (NPs) are exposed to an array of biomolecules that form a corona around the NPs, which significantly alters the surface properties of the NPs. HANPs have a rod-like shape, which prefers particle growth along the c-axis with a strong inhomogeneous electron distribution and a high surface area. The morphology of HANPs has a significant effect on regulating ion release and further controls the HANP interaction with protein (Pepsin A). For example, Kadu (2021) et al. revealed the effect of four different HANP morphologies and subsequent surface modification with cetylpyridinium chloride (CPC) on protein adsorption, having Cl^−^ as counter-ion [49]. The results show that the morphology of nanoparticles has a significant effect on the release of counter-ion, resulting in changes in the structural conformation of proteins that control their interactions with proteins. In addition, as the binding efficiency of modified HANPs to effective binding sites, the misfolding order of Pepsin A is short rod < long rod < spherical < cubic NPs, in which isotropic CPC has higher interaction with anisotropic NPs compared to functionalized NPs. In addition, Zhang (2015) et al. studied the ability of mesoporous hydroxyapatite (M-HA) and hydroxyapatite (HA) to adsorb proteins [50]; when these materials were soaked for a prolonged time in bovine serum albumin (BSA) solution, the adsorption amount of BSA increased to reach the saturated adsorption of M-HA due to its high specific area and mesoporous structure. They also studied the adsorption and release behavior of M-HA and HA in different pH environments. With decreasing pH value, the adsorption capacity of both groups (M-HA and HA) of BSA showed an upward trend; an alkaline pH of 8.4 resulted in greater charge repulsion between BSA and particles, which is not conducive to adsorption. In the subsequent release test, the BSA release duration of M-HA and HA in the alkaline environment was longer than that in the neutral/acidic environment, and the initial burst release slowed down. He (2015) et al. compared the BSA adsorption and release behavior of mesoporous HANPs (M-HANPs) and solid HANPs [51] and found that the BSA loading in M-HANPs (182 mg/g) was greater than in solid HANPs (102 mg/g) because of M-HANPs having higher specific surface area and larger pore volume than solid HANPs. By contrast, solid HANPs have no internal mesoporous structure, so BSA is mostly adsorbed on the outer surface, resulting in limited load-carrying capacity. In the adsorption mechanism of apatite, intermolecular forces, such as van der Waals, electrostatic, hydrogen bonding, and hydrophobic attractions, mainly lead to adsorb proteins. BSA has a better adsorption affinity for Ca^2^^+^ sites on the surface of apatite. The amount of BSA adsorbed on the surface of solid HANPs is less than that of M-HANPs, and the absorption is effectively increased through the additional mesoporous structure. In the release test, M-HANPs will limit the outward diffusion of the drug due to the mesoporous structure, thereby prolonging its release time. Figure 1 shows that the bioactive surface of HANPs mimics the surface of natural bone to facilitate more protein adsorption through pore adsorption, surface charge adsorption, and covalent adsorption of ions by effectively controlling drug release through pH changes in the implanted environment.

### 2.2. Effect of HA Structure on Drug Adsorption

The biological properties of biomaterials are greatly affected by their protein adsorption properties, which are related to the structures and properties of biomaterials and proteins. HA nanoparticles with a mesoporous structure (M-HANPs) could be an ideal drug carrier due to its physicochemical properties, good bioactivity, and bioabsorption. Templated hydrothermal synthesis is the most common and easiest method used to prepare M-HAs [53,54,55,56,57,58,59,60]. In contrast to solid HANPs, the hollow mesopores in M-HANPs possess numerous pore structures, which help increase the specific surface area for more efficient adsorption of chemotherapeutic drugs via electrostatic interactions. In addition, the hollow mesopores in M-HANPs can respond to changes in pH and burst release in the initial stage under acidic conditions, which limits their application in the field of drug delivery [53]. The inner space of M-HANPs contains a large number of voids, which can serve as drug storage sites; at the same time, the hollow shell acts as a permeation barrier to limit the burst release of drugs [28,54,55,56]. Munir et al. (2018) compared the effect of hollow and solid M-HANPs loaded with ciprofloxacin (CFC) [57]. The pore size distribution of the hollow M-HANP is narrow, about 3.6 nm, while the pore size distribution of the solid M-HANP is wider, about 22.58 nm. The specific surface area difference between hollow and solid M-HANPs was nearly 16 times, and the kinetics of CFC release was of zero-order. The results reconfirmed that the large specific surface area and porous structure of M-HANP resulted in the high drug loading capacity, which could effectively improve the drug utilization. Safi et al. (2018) synthesized M-HANPs with CaCO_3_ through a low-temperature solvent method and explored their ability to carry ibuprofen (IBU) [58]. Micropores and mesopores coexist in the HA structure with a specific surface area of 85 m^2^/g, which is more conducive to drug adsorption. The initial release of IBU was relatively low within 5–180 min and peaked after 4 h, which was complemented by the release of IBU adsorbed on the HA surface followed by the release in the pores of HA; hence, the porous structure can significantly prolong the drug release time, and the release rate is about 50% IBU impregnation after 12 h of release. Chen (2020) developed hollow hierarchical M-HA/poly(N-isopropylacrylamide- co-acrylic acid) and gold nanorods through electrostatic self-assembly for multi-stimuli remotely controlled drug delivery to evaluate the benefits of carrying doxorubicin hydrochloride (DOX) [53]. The results showed excellent sustained release and multiple responsive release characteristics under near-infrared light and pH 4.5. This controllable intelligent drug carrier can be applied to different drug delivery routes and has a good prospect in the field of photothermal chemotherapy. Figure 2 shows the possible mechanisms of drug adsorption by carboxylic acid to the surface-modified HANPs, and the drug is easily carried for the electrostatic attraction of drug and modified HANPs.

Benedini (2019) et al. investigated the adsorption and release of HA by using two active drugs, namely, ciprofloxacin (CIP) and IBU. The charged CIP molecules mainly interact with Ca^2^^+^ and PO_4_^3−^ in HA to achieve adsorption effect; IBU produces electrostatic adsorption with HA modified by amino acid L-arginine. CIP had the highest adsorption concentration at pH 6, but the release percentage was the lowest; while IBU had the highest adsorption concentration at pH 7.4, and the release percentage was the lowest at pH 6. The adsorption kinetics of both drugs belonged to the Avrami’s model [36]. Lee (2010) et al. studied the effect of charge functionalization of the HA surface on the loading of curcumin by characterization of carboxylic acid-functionalized HA (CA-HA) [59]. The HA crystal length shortened with increasing number of carboxylic groups, similar to the report of Ishihara (2019) et al. [60]; that is, the strong interaction between acidic groups and Ca^2^^+^ reduced the free concentration of Ca^2^^+^ during HA formation, resulting in shrinking of HA crystals and decreasing particle crystallinity. The electrostatic interaction between curcumin particles and CA-HA occurs mainly through opposite charges, and CA-HA exhibits better anti-cancer effect. Researchers still face challenges in processing polymer/NP composites to convert into nanofibers to maximize their practical applicability. Table 1 summarizes the differences in the structures of HANPs with and without mesopores as templates in carrying proteins/drugs for biomedical applications.

## 3. Recent Strategies for Compounding Natural and Synthetic Polymers with HA

Regenerative tissue is composed of multiple proteins and polysaccharides that assemble into an organized network that provides structural support to cells. Natural polymers, e.g., collagen, cellulose, gelatin, silk fibroin, keratin, chitosan, alginate, etc., are commonly used for scaffolds and have the potential advantage of supporting cell adhesion and function [61,62,63,64]. The diversity of HA/polymer composites as scaffold materials has been driven by the structural composition and function of polymers as well as immunogenicity and pathogen transmission. This review summarizes the most commonly used HA (especially for HANPs)/polymer composites for biomedical applications.

### 3.1. Electrospun Composites of HANPs/Organics

Electrospinning is a direct, inexpensive, and unique method for producing novel fibers with diameters of 100 nm and even smaller [65,66,67,68,69,70,71,72,73,74]. In fibrous membranes, polymer solutions, suspensions of HANPs containing drug and protein additives are electrospun in an electric field (Figure 3), since electrospinning is a simple, convenient, and low-cost nanofiber fabrication technique. Generally, a high-voltage electric field is used to pull the electrospinning colloidal solution containing organic solutes, vaporized solvents and HANPs additives in the syringe into fibers, so that the solvent is completely volatilized between the needle and the collector to obtain a HANPs composite fibrous membrane [65,66,67,68]. To achieve the desired viscosity for electrospinning, and to adjust the voltage, flow rate, and spinning distance to prepare HANPs composite fiber membranes, it is necessary to carefully tune the dispersed HANPs in the polymer matrix [69].

Watcharajittanont (2020) et al. electrospun TiO_2_, HANPs, and polyurethane (PU) into fibrous membranes for maxillofacial and oral surgery [70]. Sani (2021) et al. incorporated different concentrations of HANPs into chitosan (CS) and poly (ε-caprolactone) (PCL)/CS graft copolymers to fabricate bone-like fibrous scaffolds [71]. Wang (2021) et al. added HANPs into CS/gelatin to form reinforced-polyelectrolyte complex nanofibers as encapsulation for controlled release of tetracycline hydrochloride (TCH) [72]. Chuan (2020) et al. prepared a composite stereoscopic nanofiber membrane through electrospinning using a poly(lactic acid) (PLA) matrix and uniformly dispersed HANPs [73]. Chen (2019) et al. introduced coaxial electrospinning technology to prepare HANPs/gelatin-chitosan core–shell nanofiber for biomimetic composite scaffolds [74].

### 3.2. 3D Printing of Scaffolds for Tissue Engineering

3D printing technology follows the additive principle, that is, point by point and layer by layer, to create solid objects by computer-aided modeling (Figure 4). Compared with traditional production methods, 3D printing has the advantages of fast molding speed, high precision, and suitability for producing complex shapes. In vivo application, irregular defect sections and intricacies of simulated tissue can be assembled by precise positioning [75,76,77]. The HA granules smaller than the nozzle size are mixed with polymers and polymerized by adding a binder or by curing; the printing parameters (shape, size, pores, etc.) are set to prepare a 3D structural scaffold for the desired application.

Iglesias-Mejuto (2021) et al. prepared 3D-printed alginate aerogel scaffolds containing HANPs through combination of 3D printing and supercritical CO_2_ drying for bone regeneration [78]. Their results showed that HANPs and CaCl_2_ (the major provider of Ca^2+^ concentration) determined the scaffold texture. Cestari et al. (2021) fabricated a composite of bioderived PCL and HANPs by 3D printing to obtain porous scaffolds for bone regeneration [79]. Wei (2021) et al. used 3D-printed HA microspheres enhanced with poly (lactic-co-glycolic acid) (PLGA) to evaluate the efficiency for bone regeneration scaffolds [80]. Chen (2019) et al. investigated the 3D printing of composite scaffolds composed of HA and gelatin, CS, and carboxymethyl cellulose (CMC) [81]. Yeo (2021) et al. studied 3D-printed poly(glycolic acid)/HA composite scaffolds to promote bone regeneration [82]. The most important advantage of 3D-printed scaffolds is that 3D scaffolds can be used as tissue models to replicate the structural complexity of living tissues. Therefore, not only the biomaterials used but also the macroscopic, microscopic, and nanostructures of scaffolds are crucial. The 3D-printed bone scaffold containing HANPs in a biopolymer-based bio-ink formulation may provide a viable option for promoting patient specific tissue regeneration through precisely control of scaffold structure and composition.

### 3.3. Freeze Drying to Prepare Scaffolds

Freeze drying is the best drying technique for heat-sensitive food materials over other conventional drying techniques. During the process, ice evaporates directly without forming a liquid phase (sublimation) due to reduced pressure (Figure 5). However, freeze drying is an expensive and time-consuming technique, which limits its use in drying heat-sensitive and high-value products. The porous scaffold obtained by this method has the characteristics of high porosity and interconnected pores, but the pore distribution is relatively uneven due to the size and distribution of ice crystals. During freeze drying, the solvent initially solidifies, allowing the polymer and HA to enter the interstitial spaces. The frozen mixture is then lyophilized using a freeze dryer, where the ice solvent is evaporated [30,83].

Ma (2021) et al. prepared biomimetic gelatin/chitosan/polyvinyl alcohol/HANP scaffolds for bone tissue engineering by freeze drying [84]. Xing (2021) et al. prepared chitin–hydroxyapatite–collagen composite scaffolds for bone regeneration by freeze-drying [85]. Pottathara (2021) et al. [86] prepared gelatine/collagen/HANP scaffolds by unidirectional freeze-casting. Kane (2012) et al. investigated the effect of HA addition and morphology on the structure and compressive mechanical properties of freeze-dried collagen scaffolds [87]. Brahimi (2022) et al. prepared highly porous chitosan/HA scaffolds through freeze gelation by varying the HA content [88]. Feroz (2021) et al. developed a novel hydroxypropylmethyl cellulose (HPMC) crosslinked keratin scaffold with HA as the main inorganic component for alveolar bone regeneration by freeze-drying technology [89]. By incorporating HANPs into the polymer matrix of dextran/chitosan, El-Meliegy (2018) et al. realized a novel composite scaffold by freeze-drying technique and determined the effect of HANPs on scaffold morphology and mechanical properties [90]. They found the presence of HANP as a reinforcement can noticeably enhance the elastic modulus and compressive strength of the HANPs composite scaffolds.

### 3.4. Other Techniques

All of the above properties enhance the technology of HANPs/polymers for biomedical applications, and many other strategies are available. For example, Nabavinia (2019) et al. investigated the effects of HANPs–alginate–gelatin-based microcapsules as a cell adhesion molecule and HANPs as an osteoconductive component on the properties of alginate-based hydrogels and evaluated the behavior of microcapsule osteoblast-like cells by using factorial experimental design technique [91]. Silva (2022) et al. processed polyvinylidene fluoride (PVDF) and HA composite filaments by twin-screw extrusion with different processing screw fin angular speeds [92]. Wenzhi (2021) et al. prepared microspheres of HANPs and poly (lactide-co-glycolide) nanocomposite for bone repair by a novel airflow shearing technology and evaluated its potential for clinical application as in vivo bone repair fillers [93]. Mahmoud (2020) et al. produced alginate/HANPs composite scaffolds by utilizing fish bones as a biosource for HANPs [94]. Their 3D porous scaffolds were fabricated using a sponge polymeric approach and then coated with alginate to enhance biodegradability and osteoconductivity.

## 4. Polymers–HA Composite as Carriers for Drug-Sustained Release

In summary, HA has the characteristics of adsorbing drugs and biocompatibility and is an ideal drug carrier material. However, the initial release rate of the drug is very fast due to the weak interaction between the drug and the HA particles. In addition to its excellent mechanical properties and surface functionality, polymers–HA composite can be used to prolong drug release, making HA/polymer composites suitable as carriers for drug-sustained release [95]. Different carrying targets (such as antibiotics, anti-inflammatory, and anti-cancer drugs, natural extracts, growth factors, etc.) can be added to achieve the local and special needs for clinical surgical treatment. For example, in Figure 6, HANPs/polymer composites of different structures can be used as carriers of drugs, proteins, or antibacterial agents.

### 4.1. Membrane Form

Eskitoros-Togay (2020) et al. incorporated different ratios of HANPs and curcumin into the same PCL/poly (ethylene oxide) (PCL/PEO) mixed matrix to form membranes by electrospinning. The encapsulation efficiency of curcumin in the electrospun membranes was 86–94%. The fiber membrane containing 0.3% HANPs showed a gradual increasing trend in the first hour of release and only reached 43% of curcumin released from the release time to the eighth hour, indicating that the controlled release of curcumin can be achieved in a simulated environment to prolong the action of the drug after implantation [96]. By varying the weight ratio of sodium alginate and gelatin (A/G = 40/60, 50/50, and 60/40) and adding different concentrations of HANPs (1, 2, 5, 10, and 20% *w*/*w*) to the film solution, Türe (2019) et al. explored whether the addition of HANPs alters the physical, mechanical, thermal, and antimicrobial properties of the films. In addition, tetracycline hydrochloride (TH) was chosen as a model to study drug release in water. Their results showed that the swelling rate and weight loss decreased as the amount of alginate. Membrane structures with high alginate content were denser compared to the increased amount of HA that resulted in rougher surfaces. Films with lower tensile and elastic modulus values with greater than 1% HANPs for A/G = 50/50 and 60/40 [97]. The amount of TH released decreased with increasing amount of HA, as the addition of HA acted as a barrier and reduced the drug release. In this study, swelling behavior and TH release have similar patterns. Prakash (2019) et al. fabricated HANPs-incorporated polyvinyl alcohol-sodium alginate (PVA-SA) membranes for controlled release of the antibiotic amoxicillin to treat subosseous periodontal defects [98]. In this study, the polymer tends to degrade, leading to drug release, and SA dissolves faster in aqueous systems compared with PVA at room temperature; therefore, when SA degrades in the polymer matrix, the drug molecules tend to detach from the membrane and acts on the infected area. The result showed that the amount of amoxicillin released 43% of the drug on day 3, 72% on day 6, and 87% on day 10, suggesting that the drug release from these composite membranes were sustained. Ramírez-Agudelo (2018) et al. released doxycycline (Dox) and HANPs from biodegradable polymer composite nanofibers of PCL/gelatin for local drug delivery [99]. Dox and HANPs were encapsulated at various PCL/Gel ratios (70:30, 60:40, 50:50 wt%). They prepared Dox/HANPs-loaded PCL-Gel composite fibers by electrospinning. The release kinetics of Dox can be shown in two phases: in the first phase, all scaffolds exhibited about 60% burst effect release in the first hour; in the second release stage, the remaining loaded drug can be released within 55 h. Baldino (2018) et al. studied the silver-loaded HANPs being incorporated into PVA membranes obtained by supercritical CO_2_ (SC-CO_2_) assisted phase inversion [100]. Their results show that HA-Ag NPs loaded in PVA membranes were more active than the HANPs alone. The bactericidal results show that the Ag^+^ concentration in the HA-Ag NPs can be reduced from 22 ppm to 11 ppm, which has a bacteriostatic effect on *E. coli*, and the Ag^+^ in the composite membranes can prolong and control its release behavior, which can be used in biomedicine, coating and filter applications.

### 4.2. Scaffold Form

Kim (2004) et al. prepared HA porous scaffolds by polymeric reticulate method with HA-PCL composite after embedding with the antibiotic drug TCH [101]. The HA/PCL ratio had a strong effect on release. In a short period (<2 h), about 20–30% of the drug was released. However, the release rate persisted for a long time and was highly dependent on the extent of coating HA/PCL dissolution. Zhang (2021) et al. prepared 3D scaffolds by using PCL/PEO/HA with different HA concentrations by direct ink writing (DIW) [102]. Vancomycin (VAN) was incorporated into composite scaffolds, and the drug release properties of the composites were investigated. The release kinetics of PCL/PEO/HA samples containing low and high VAN loadings (3% and 9% *w*/*w*) were determined in vitro. The VAN-loaded PCL/PEO/HA scaffolds exhibited a first-order immediate release, i.e., a large burst of release within a half hour. In addition to drug concentration and drug–polymer interactions, scaffolds with higher porosity and surface/volume ratio have faster VAN release profile, which can be used for hemostasis and anti-inflammation related to bone tissue applications. Martínez-Vázquez (2015) et al. fabricated Si-doped HA/gelatin porous scaffolds at moderate temperatures by using a rapid prototyping system and analyzed the release profiles of vancomycin from different scaffolds [103]. Less than 30% of vancomycin was released from the scaffold within 1 h, and the antibiotic release continued for 8 h. Hu (2014) et al. used the anti-inflammatory drug IBU to study the drug release behavior of HA/poly(l-lactic acid) (PLLA) nanocomposite scaffolds [104]. The IBU loaded in the nanocomposite exhibited a sustained release curve, and the release kinetics followed the Higuchi model with a diffusion process. Zhang (2012) et al. prepared a novel vancomycin (VCM)-loaded M-HA/CS composite scaffold by freeze-drying and used it for in vivo drug release and antibacterial studies [105]. The results indicated that drug-VCM loaded M-HA/CS composites can release VCM for a long time after implantation in vivo and exhibit effective antibacterial activity against *methicillin-resistant Staphylococcus aureus* (MRSA). Liang (2021) et al. prepared a composite hydrogel scaffold of HA and sodium alginate (SA) by using 3D printing [106]. Additives naringin (NG) and calcitonin gene-related peptide (CGRP) were used as osteogenic factors for the fabrication of drug-loaded scaffolds. The HA/SA scaffold can continuously and stably release NG and CGRP until the 21st day, and the drug was mainly released from the large pores of the scaffold to achieve the purpose of bone regeneration.

### 4.3. Spherical Form

Bi (2019) et al. investigated the drug loading behavior on HANPs, HANPs/ SA, and HANPs/SA/CS composite [107] and the drug release behavior of doxorubicin hydrochloride (Dox·HCl)-loaded HANPs/SA/CS microspheres for drug delivery system in phosphate-buffered saline (PBS) solutions with different pH values (pH 7.4, 6.5, and 5.0). The drug loading and encapsulation efficiency of HANPs were 5.9% ± 0.6% and 11.8% ± 1.2%, respectively, which were lower than those of the composite microspheres. This finding might be attributed to the weak electrostatic interaction between HANPs and drug molecules. The 3D network structure of the microspheres helps to load more drugs because the drug molecules are not only absorbed into the 3D network structure of the microspheres but also on the surface of the microspheres. However, the drug loading of HANPs/SA microspheres is higher than that of HANPs/SA/CS microspheres because their surface is coated with a thin film; the film may become an obstacle for drug molecules to be absorbed into the internal structure of the microspheres. The Dox-loaded microspheres showed faster drug release properties with decreasing environment pH. HA and CS were degradable in acidic media, and their different release behavior types were attributed to the different levels of degradability under various medium conditions. Dox-loaded microspheres adopt slow-release behavior, which can be attributed not only to electrostatic interactions between hydroxyl and amine groups but also to the uniform dispersion of HA nanoparticles inside the microspheres and on the surface. Xue (2018) et al. prepared polyvinyl alcohol (PVA)/HA composite microspheres with different HA contents by in situ synthesis and the synergistic effect of emulsification–crosslinking [108]. In this study, vancomycin hydrochloride (VH) was selected as a model drug to embed in microspheres by immersion. The PVA microspheres have larger swelling degree after proper crosslinking with glutaraldehyde through hydrogen bonding or ionic interactions and have good drug VH encapsulation efficiency and VH loading capacity. He (2021) et al. designed and fabricated microspheres and scaffolds assembled using HA and vancomycin hydrochloride (VH)-loaded polytrimethylene carbonate (PTMC) and PLLA core/shell microspheres [109]. Combining drugs, active growth substances, and CS in the same microsphere carrier provides sustained long-term release. The drug loading (original encapsulated drug = 2.6 mg) and drug loading efficiency (200 mg) of VH in the microspheres were 1.52 ± 0.06 mg and 58.3% ± 3.9%, respectively. Zhang (2010) et al. prepared pH-sensitive SA/HA nanocomposite beads in sol–gel as drug delivery vehicles, with iclofenac sodium (DS) as a model drug. Factors affecting the swelling behavior, drug loading, and controlled release behavior of SA/HA nanocomposite microspheres were studied in detail. Compared with pure SA hydrogel beads, the SA/HA-DS nanocomposite beads prepared under optimal conditions can prolong the release of DS up to 8 h. Calasans-Maia (2019) et al. prepared alginate-encapsulated nanobicarbonate hydroxyapatite (CHA) microspheres for topical delivery of minocycline (MINO) to inhibit the growth of Enterococcus faecalis [110]. The amount of MINO adsorbed on the CHA powder depends on the initial concentration of MINO in the solution, the quality of CHA, and the pH of the solution. Adsorption experiments were performed using 1.5 mg/mL MINO in PBS buffer solution (pH = 7.4) containing 50 mg/mL CHA powder. After 24 h, the amount of antibiotic loaded on the CHA powder was 25.1 ± 2.2 µg MINO/mg CHA. The MINO loss of CHA during microsphere processing was about 40%. The MINO release profile of CHA microspheres loaded with 15.1 ± 1.4 µg MINO/mg CHA in PBS buffer was evaluated. A rapid release of approximately 60% of the initially loaded MINO (9.1 µg MINO/mg CHA) was observed within the first 24 h, and the remaining 6.0 µg MINO/mg CHA sustained release was observed over 10 days. Padmanabhan (2018) et al. prepared core–shell nanocomposites with gum–acacia (GA) as shell and HANPs as core for drug delivery and tissue engineering and studied the drug release behavior of naringenin [111]. The uneven distribution of naringenin in the HANP matrix may result in a rapid release pattern. By contrast, in the case of GA-HA, the colloidal nature of GA helps to bind to naringenin efficiently, allowing for better distribution of the drug throughout the core–shell nanocomposites. Electrospinning, 3D printing, and freeze drying are commonly used to composite HANPs into thin films, regenerated membranes, scaffolds, microspheres, and coatings; drugs can also be loaded according to the composite structure to meet different clinical applications.

### 4.4. Coating

Prasanna (2018) et al. synthesized HANPs as nanocarriers for sustained release of the antibiotic amoxicillin for treatment of bone infections. Nanoparticles were then coated on PVA and SA in a layer-by-layer spray coating. Layer-by-layer coating of HANPs/PVA/SA resulted in sustained release of amoxicillin, which was observed for 30 days [112]. Bose (2018) et al. investigated the influence of PCL coating on alendronate drug release kinetics in vitro [113]. The results showed that PCL coating would minimize the burst release of alendronate from plasma-sprayed Mg-doped HA coated commercially with pure titanium (cp-Ti). In the absence of PCL coating, about 75% of alendronate was released within the first 12 h. The samples with 2 and 4 wt% PCL coatings exhibited slower burst release, with values of 34% and 26%, respectively. After 24 h, the samples without PCL coating released >75% alendronate. The samples with PCL coating released about 50% alendronate. Kong (2016) et al. used the anticancer drug DOX as a model drug to load into the resulting HANPs-polyethyleneimine (PEI)-hyaluronic acid (HyA) nanoparticles [114]. DOX@HANPs-PEI-HyA nanoparticles were subjected to buffer solutions with different pH values at 37 °C to examine the DOX release behavior in vitro. The cumulative DOX release percentage of DOX@HANPs-PEI-HyA at pH 7.4 was only 23% over 48 h, while more than 88% DOX release was observed at pH 5.0. Under acidic conditions, the amino group of DOX was protonated, which weakened the adsorption between HANPs and DOX. Meanwhile, PEI was swollen in the acidic environment due to the protonation of amine groups. Therefore, these reasons strongly promote DOX diffusion out of DOX@HANPs-PEI-HyA under acidic conditions. Therefore, this pH-responsive drug release property of DOX@HANPs-PEI-HyA nanoparticles makes it promising for cancer therapy due to the weak acidity of tumor tissues and tumor cells. Numerous studies have highlighted the role of HANPs in promoting the regeneration of tissues with polymers, and Table 2 briefly summarizes the different polymer-HA composite morphologies mentioned above.

## 5. Conclusions

Apparently, the large number of pores in M-HA can effectively increase the drug loading and slow down the initial burst release. In addition, surface modification can effectively improve the loading effect of HA on specific active molecules, further increasing the versatility of HA. The preparation strategy of HANPs composite polymer will be selected according to different clinical requirements and functions to achieve multifunctional purpose (such as wound dressing, bone tissue engineering, etc.). When the HA composite polymer is loaded with proteins/drugs, the release amount and rate are also different due to different structures and contact environments. For example, the 2D contact area of the membrane/film is lower than that of the 3D scaffold and microspheres. Drugs released more easily due to the inherent porosity of the microspheres due to the larger specific surface area and the expansion of the scaffold after implantation. In addition, the degradation rate of the polymers, the pH value of the drug release environment, the characteristics of the drug itself, such as lipid (fat)-soluble or water-soluble, and the bond between the drug and HA and macromolecules, affect the release. The protein/drug-carrying HANPs have large changes in physical and chemical properties due to changes in surface conditions, which will affect their application ability as carriers to introduce functions into repair sites. Therefore, further selection of biomolecules, proteins/drugs, preparation methods and structures is required to prepare HA-templated carriers.

## Figures and Tables

**Figure 1 polymers-14-00976-f001:**
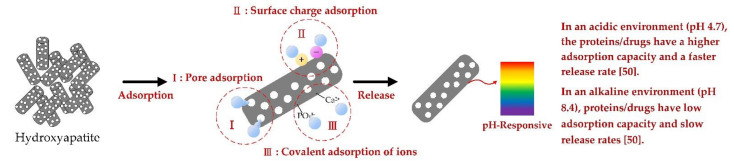
Schematic diagram of the adsorption and attachment factor mechanism of porous HA nanorod [52].

**Figure 2 polymers-14-00976-f002:**
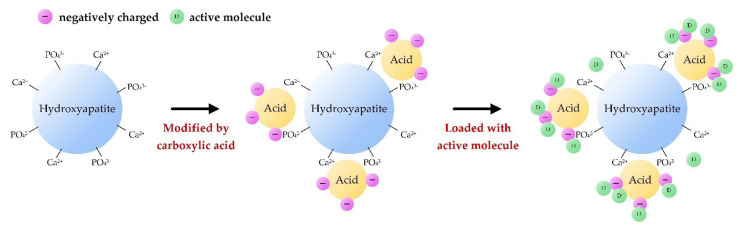
Schematic diagram of drug adsorption on the surface-modified HANPs. (−: negatively charged biding sites on surfaces of modified HANPs; D: active molecule of drugs.)

**Figure 3 polymers-14-00976-f003:**
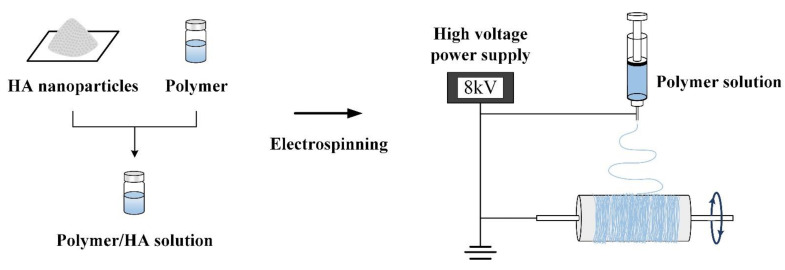
Fiber-based membranes were prepared by electrospinning simulated polymer composite HANPs.

**Figure 4 polymers-14-00976-f004:**
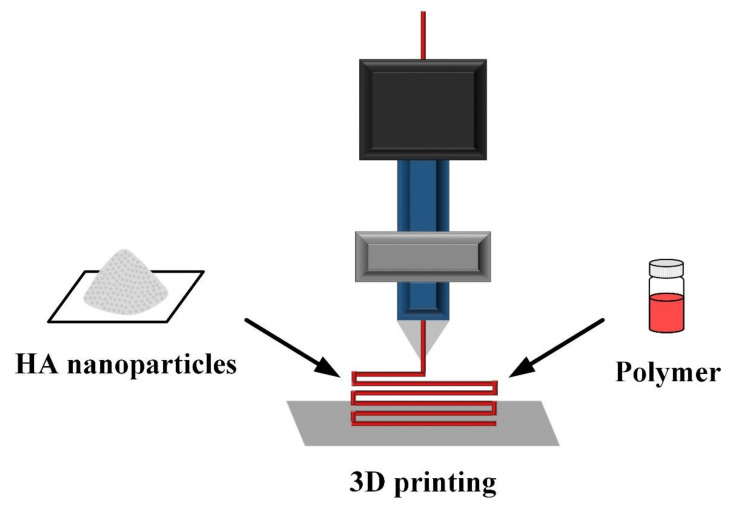
Schematic diagram of 3D printing to prepare polymer composite HANP scaffolds.

**Figure 5 polymers-14-00976-f005:**
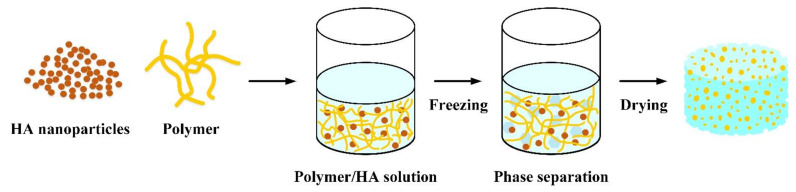
Schematic diagram of the preparation of porous scaffolds of polymers composite HANPs prepared by freeze-drying.

**Figure 6 polymers-14-00976-f006:**
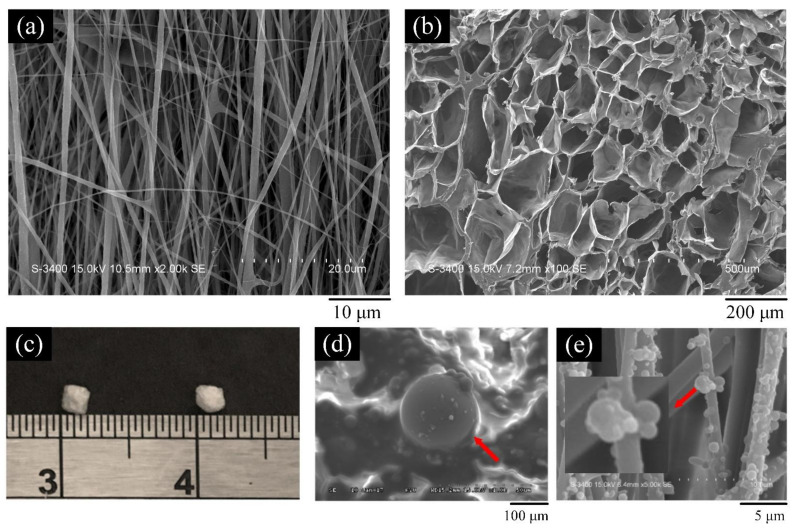
Appearances of (**a**) electrospun film; (**b**) freeze-dried scaffold; (**c**) freeze-dried beads (**d**) microspheres, and (**e**) electrospray nanospheres.

**Table 1 polymers-14-00976-t001:** Related applications of different types of HANP structures as a template in carrying proteins/drugs.

HANP Structures	Proteins/Drug	Highlights and Potential Clinical Applications	Ref.
Solid (non-porous) hydroxyapatite nanoparticles (HANPs)	Pepsin A	Comparing the effects of different types of HA modified with cetyl pyridine chloride on the interaction with pepsin A, HANPs have higher enzymatic activity (18.45%) than microscale. HANPs with surface modification can improve their use in biomedical applications potential.	[49]
Mesoporous hydroxyapatite nanoparticles (M-HANPs)	Bovine serum albumin (BSA)	The adsorption capacity of M-HANPs in acidic environment (pH 4.7) was higher than that of micro-HA particles. In alkaline environments (pH 8.4), they have smaller bursts and flatter release profiles, which can be used for targeted drug delivery and bone therapy.	[50]
Mesoporous hydroxyapatite rod-like nanocrystals	Fetuin from serum protein	Fetuin has the ability to inhibit the growth of M-HA nanocrystals to form dumbbell shaped, mesoporous structure, and large surface area. M-HAs of rod-like crystal size (235–515 nm) with inner mesopores (21–31 nm) can load more drugs and sustained-release drugs, which is beneficial to the field of drug delivery and sustained-release as drug delivery vehicles.	[51]
Hollow mesoporous hydroxyapatite nanoparticles	Doxorubicin (DOX)	The hollow mesoporous structure of M-HANPs has high biocompatibility and good drug loading capacity, the drug loading rate is increased from 17.9% to 93.7%, and has excellent drug nanocarrier performance as carriers of large pharmaceutics.	[54]
Solid and mesoporous hydroxyapatite nanoparticles	Ciprofloxacin	Compared with solid HANPs, M-HANPs have higher specific surface area and high drug loading, and have greater application potential in the field of drug delivery. Therefore, M-HANPs can potentially be used in smart drug delivery systems.	[57]
Functionalization of hydroxyapatite nanoparticles	curcumin nanoparticles	Carboxylic acid surface modification of HANPs can enhance the adsorption of curcumin and improve its drug availability. Curcumin-modified HANPs have better anticancer activity and have good potential in the field of medical regeneration.	[60]

**Table 2 polymers-14-00976-t002:** Characteristics and possible clinical applications of different types of polymer-HA composites.

Biomolecules with Different Types of Appearance	Drug	Highlights and Potential Clinical Applications	Ref.
TCH/HANPs/CG core–shell nanofibers	Tetracycline hydrochloride (TCH)	The composite nanofibers have long-lasting antibacterial function, good biocompatibility, and high mechanical strength, and are suitable for wound dressings and drug delivery systems.	[72]
HANPs/PLGA microspheres	−	The diameter of the composite microspheres is about 250 μm. When the content of HANPs was 20% and 40%, respectively, it could promote the mineralization and osteogenic differentiation of MC3T3-E1 cells, and had good clinical application potential in bone tissue engineering and bone implantation.	[93]
HANPs-containing alginate–gelatin composite films	Tetracycline hydrochloride (TCH)	The addition of HANPs will make the surface of the composite film rougher and effectively improve the thermal stability. In addition, it can reduce the initial burst release of the drug. The polymer-HA composite film can be used not only for biomedical applications, but also for food packaging.	[97]
Polycaprolactone/ polyethylene oxide/ hydroxyapatite 3D scaffolds	Vancomycin (VCM)	The composite scaffold with HA content of 65% had the best wettability and mechanical properties, but adding too much HA would affect the mechanical properties of the polymer-HA composite. The drug release showed an initial burst, and the 3D scaffold with antibacterial activity was suitable for bone tissue engineering applications.	[102]
A chitosan (CS)-coated polytrimethylene carbonate (PTMC)/polylactic acid (PLLA)/oleic acid-modified hydroxyapatite (OA-HA)/vancomycin hydrochloride (VH) microsphere scaffold	vancomycin hydrochloride (VH)	Two active molecules, OA-HA and VH, can be released through the pores. In addition to facilitating osteoblast adhesion, CS coating can also control the release behavior of the OA-HA to stimulate the proliferation of osteoblasts, which is expected to be used in bone tissue engineering.	[109]

## Data Availability

The data presented in this study are available on request from the corresponding author.

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
