# Peer review of "Advances of Hydroxyapatite Hybrid Organic Composite Used as Drug or Protein Carriers for Biomedical Applications: A Review"

_polymers, 2022, doi:10.3390/polym14050976_

Round 1
Reviewer 1 Report
The manuscript written by Huang and collaborators presents the application of hydroxyapatite-polymer composite for medical treatment. The topic is timely and very interesting. However, in the present form, the manuscript is difficult to read and in some parts a bit boring. The reason is that the text is not mature enough, too descriptive with many vague sentence. Actually, it is more an overview than a review because many data are brought without critical analysis. Many information could be summarized in tables. Also, there is insufficient information on whether these materials have been tested in vivo. In my opinion to make the text more interesting, the data should be compared a discussed deeper than they are in the present manuscript.
Minor changes are listed below:
Line 33, There should be „polymer-bioceramic composites” instead of “bioceramics composite polymers”.
Line 47, What does “The target adsorption capacity of HANPs (…)” mean?
Line 50. There should be “drug loading capacity, drug delivery efficiency, (…)” instead of “drug delivery capacity, efficiency, (…)”.
Line 51. It is not clear whether the surface charge of HANPs can control protein release or protein biding?
Lines 55-56. I do not understand the statement “(…) the adsorption relationship with proteins (…)”
Lines 56-58. This a repetition, please removed it.
Lines 82-86. I do not understand.
Line 92. The authors, by "both groups", meant nanoparticles? It is not clear.
Line 93. I guess the authors meant electrostatic repulsion between oppositely charged particles and protein molecules. Please give the range of the pH, what does mean “higher pH”?
Line 98. “(…) the BSA drug loading of M-HANPs was 182 mg/g greater than 102 mg/g in solid HANPs.” It would be better to give the values in parentheses. In that form it is not readable.
Line 105. The space is missing.
Line 115. On the Fig. 1, the authors present that the releasing of the therapeutic/protein from the HA carrier can be by pH-changing, light and thermal exposure. However, this was not mentioned in the text. How is it done? Also, there is no information on the pH range, at which HA can dissolve and release the drug.
Line 125. I guess the authors meant “via” electrostatic interactions not “in”.
Line 136. What does “impregnation concentration of the drug” mean?
Lines 184-230. Please check the text formatting.
Lines 217-219. I do not understand.
Line 281. There should be “polymers-HA composite” .
Lines 305-308. Have the composite properties been improved by changing the weight ratio of polymer to HA?
Line 353. The name of bacteria should be written in italic. Please check it throughout the manuscript.
Line 372. Please remove “drug”.
Author Response
Detailed responses to Reviewer’s comments
Reviewer #1
The manuscript written by Huang and collaborators presents the application of hydroxyapatite-polymer composite for medical treatment. The topic is timely and very interesting. However, in the present form, the manuscript is difficult to read and in some parts a bit boring. The reason is that the text is not mature enough, too descriptive with many vague sentence. Actually, it is more an overview than a review because many data are brought without critical analysis. Many information could be summarized in tables. Also, there is insufficient information on whether these materials have been tested in vivo. In my opinion to make the text more interesting, the data should be compared a discussed deeper than they are in the present manuscript.
Response: We greatly appreciate the Reviewer’s efforts that helped improve the manuscript. All comments have been addressed and appropriate revisions have been made to the entire manuscript, in particular we have summarized the 2 tables, accordingly. We hope that the following modifications are acceptable:
Minor changes are listed below:
Line 33, There should be „polymer-bioceramic composites” instead of “bioceramics composite polymers”.
Response: Thanks for the correction, we changed it (Line 40, revised manuscript)
Line 47, What does “The target adsorption capacity of HANPs (…)” mean?
Response: Thanks for the correction, we removed the incorrect expression. (Line 54, revised manuscript)
Line 50. There should be “drug loading capacity, drug delivery efficiency, (…)” instead of “drug delivery capacity, efficiency, (…)”.
Response: Thanks for the correction, we have adopted the suggestion. (Lines 57-58, revised manuscript)
Line 51. It is not clear whether the surface charge of HANPs can control protein release or protein biding?; Lines 55-56. I do not understand the statement “(…) the adsorption relationship with proteins (…)”; Lines 56-58. This a repetition, please removed it.
Responses: Thank you, we have adopted the above suggestions and made appropriate changes to the expression (Lines 58-71, tracking changes in revision).
Lines 82-86. I do not understand.
Response: I have rewritten the expression, hopefully the changes are acceptable. (Lines 96-106, tracking changes in revision)
Line 92. The authors, by "both groups", meant nanoparticles? It is not clear.
Response: I have explain the both groups, porous hydroxyapatite (M-HA) and hydroxyapatite (HA). (Line 112, tracking changes in revision).
Line 93. I guess the authors meant electrostatic repulsion between oppositely charged particles and protein molecules. Please give the range of the pH, what does mean “higher pH”?
Response: Thanks, the exact pH has been provided (Lines 112-114, tracking changes in revision).
Line 98. “(…) the BSA drug loading of M-HANPs was 182 mg/g greater than 102 mg/g in solid HANPs.” It would be better to give the values in parentheses. In that form it is not readable.
Response: Thanks, we heeled the suggestions and the values has been provided in parentheses. (Lines 118-120, tracking changes in revision)
Line 105. The space is missing.
Response: I would like to thank the reviewer for the detailed review, and I have resolved this issue. (Line 126, tracking changes in revision)
Line 115. On the Fig. 1, the authors present that the releasing of the therapeutic/protein from the HA carrier can be by pH-changing, light and thermal exposure. However, this was not mentioned in the text. How is it done? Also, there is no information on the pH range, at which HA can dissolve and release the drug.
Response: I would like to thank the reviewers for very constructive suggestion. The Figure 1 has been updated. (Line 135, tracking changes in revision)
Line 125. I guess the authors meant “via” electrostatic interactions not “in”.
Response: Thanks, we have changed it. (Line 147, tracking changes in revision)
Line 136. What does “impregnation concentration of the drug” mean?
Response: Thanks, we have changed the improper expression. (Lines 158-159, tracking changes in revision)
Lines 184-230. Please check the text formatting.
Response: I have rewritten the text formatting, hopefully the changes are acceptable. (Lines 213-238, tracking changes in revision)
Lines 217-219. I do not understand.
Response: Thanks, I have rewritten the sentences, hopefully the changes are acceptable. (Lines 251-256, tracking changes in revision)
Line 281. There should be “polymers-HA composite” .
Response: Thanks, we have adopted this recommendation and made changes accordingly. (Lines, 316, and 320-321, tracking changes in revision)
Lines 305-308. Have the composite properties been improved by changing the weight ratio of polymer to HA?
Response: Thanks for your suggestion and we have provided the revised results (Lines 342-347, tracking changes in revision).
Line 353. The name of bacteria should be written in italic. Please check it throughout the manuscript.
Response: Thanks, we have changed it. (Line 399, tracking changes in revision).
Line 372. Please remove “drug”.
Response: Thanks, we have changed it. (Line 417, tracking changes in revision).
Reviewer 2 Report
The manuscript “Advances of hydroxyapatite hybrid organic composite used as drug or protein carriers for biomedical applications: A review” is a review on the use and influence of Hydroxyapatite as a carrier on the adsorption and release of active molecules. The work is clear and well organized. However, some revisions are required, as follows:
- Use the reference style in text as suggested by the Journal;
- Add the main findings/conclusions also in the Abstract;
- Add a brief description on the use of HA nanoparticles loaded in porous structures, as for instance in this case: Baldino et al., Production, characterization and testing of antibacterial PVA membranes loaded with HA-Ag3PO4 nanoparticles, produced by SC-CO2 phase inversion, Journal of Chemical Technology and Biotechnology, 2019, 94(1), pp. 98–108; etc.;
- Correct typos;
- In section 3.4., supercritical CO2 assisted processes can be cited/described for the production of loaded porous 3-D structures, like aerogels and membranes.
Author Response
Detailed responses to Reviewer’s comments
Reviewer #2
We greatly appreciate the Reviewer’s efforts that helped improve the manuscript. All comments have been addressed and appropriate revisions have been made to the entire manuscript
The manuscript “Advances of hydroxyapatite hybrid organic composite used as drug or protein carriers for biomedical applications: A review” is a review on the use and influence of Hydroxyapatite as a carrier on the adsorption and release of active molecules. The work is clear and well organized. However, some revisions are required, as follows:
- Use the reference style in text as suggested by the Journal;
Response: Thanks, we have checked and made appropriate changes.
- Add the main findings/conclusions also in the Abstract;
Response: Thanks, we have adopted this recommendation and made changes in Abstract.
- Add a brief description on the use of HA nanoparticles loaded in porous structures, as for instance in this case: Baldino et al., Production, characterization and testing of antibacterial PVA membranes loaded with HA-Ag3PO4 nanoparticles, produced by SC-CO2 phase inversion, Journal of Chemical Technology and Biotechnology, 2019, 94(1), pp. 98–108; etc.;
Response: Thanks, We've added a brief description and added the new reference you suggested.
Correct typos;
- In section 3.4., supercritical CO2 assisted processes can be cited/described for the production of loaded porous 3-D structures, like aerogels and membranes.
Response: Thanks, we have corrected it.
Round 2
Reviewer 1 Report
Many thanks to the authors for their comprehensive responses to my comments. The paper is ready to be published in the journal.
Reviewer 2 Report
The authors performed the modifications proposed by the Reviewer and improved the manuscript.